# Social Isolation Among Individuals with Incontinence: A Scoping Review

**DOI:** 10.3390/nursrep15110375

**Published:** 2025-10-24

**Authors:** Valentina Stroppa, Paolo Iovino, Ilaria Marcomini, Roberto D’Errico, Andrea Poliani, Debora Rosa, Duilio Fiorenzo Manara, Giulia Villa

**Affiliations:** 1Center for Nursing Research and Innovation, Faculty of Medicine and Surgery, Vita-Salute San Raffaele University, 20132 Milan, Italy; stroppa.valentina@hsr.it (V.S.); r.derrico@studenti.unisr.it (R.D.); poliani.andrea@unisr.it (A.P.); rosa.debora@unisr.it (D.R.); manara.duilio@hsr.it (D.F.M.); villa.giulia@hsr.it (G.V.); 2Department of Clinical Cardiology, Istituto di Ricovero e Cura a Carattere Scientifico (IRCCS) Ospedale San Raffaele, 20132 Milan, Italy; 3Health Sciences Department, University of Florence, 50139 Florence, Italy; paolo.iovino@unifi.it; 4Department of Internal Medicine, ASST Grande Ospedale Metropolitano Niguarda, 20158 Milan, Italy

**Keywords:** incontinence, loneliness, ostomy, surgical stoma, social isolation, scoping review

## Abstract

**Introduction:** Urinary and fecal incontinence, as well as the presence of an ostomy, are globally prevalent conditions with substantial implications for individuals’ daily lives. Among the psychological consequences, social isolation is a frequently reported experience but remains poorly explored in the existing literature. The aim of this scoping review is to explore how social isolation has been conceptualized and operationalized in research on individuals with incontinence and to synthesize evidence on its antecedents and outcomes. **Methods:** This review was conducted in accordance with the Joanna Briggs Institute guidelines and reported following the PRISMA-ScR checklist. Data were thematically synthesized and interpreted according to the Middle Range Theory of Social Isolation in Chronic Illness. **Results:** Twenty-three studies met the inclusion criteria. Findings indicate that social isolation among individuals with incontinence is a complex, multifactorial phenomenon. Predisposing factors included individual needs for social interaction and desire for approval, psychological resilience, toilet accessibility, education, income, gender, and age. Precipitating factors were related to illness trajectory and adaptation processes, including ostomy acceptance, time since ostomy creation or oncological treatment, sense of belonging, perceived social support, stigma, self-esteem, clinical severity, illness-related conditions, and loss of autonomy. Reported outcomes were consistently adverse, encompassing depression, anxiety, and reduced quality of life. **Conclusions:** Social isolation represents a core dimension of the lived experience of incontinence and should be recognized as a key clinical outcome. Systematic screening and targeted interventions should be integrated into continence care pathways. Future research should adopt longitudinal and interventional designs to clarify causal mechanisms and evaluate strategies to prevent and mitigate isolation.

## 1. Introduction

Incontinence refers to the involuntary loss of urine or feces, which can significantly impair daily functioning and quality of life. Urinary incontinence is defined by the International Continence Society as “any involuntary leakage of urine” [1]. Fecal incontinence encompasses the unintentional loss of solid or liquid stool or gas [2]. In some cases, individuals carry a urinary or intestinal stoma, which is a surgically created opening that bypasses normal elimination pathways due to chronic illness, injury, or cancer. The presence of an ostomy leads to a permanent change in continence function. These conditions, often considered benign, or even normal parts of the aging process, are in fact chronic health problems requiring long-term management [3].

Urinary incontinence affects more than 400 million people over the age of 20, representing more than 20% of the global adult population [4]. Fecal incontinence is less prevalent, but still significant, with global estimates ranging from 5% to 10%, depending on age, gender, and comorbidity burden [5]. Regarding ostomy, about 1 million individuals in the United States are estimated to be living with this device, which often results from colorectal cancer of inflammatory bowel disease [6].

Incontinence is associated with a range of healthcare, functional, and psychological consequences, making it a relevant public health issue. Specifically, incontinence increases the risk of falls, urinary tract infections, subsequently leading to loss of independence, institutionalization, and decreased mobility [7,8]. Psychosocial effects are also profound, with shame, embarrassment, loss of self-esteem, and social isolation being frequently reported [9].

Social isolation is a complex and multifaceted phenomenon that refers to a reduction or absence of meaningful social connections. This phenomenon is typically distinguished into objective social isolation, which involves a measurable reduction in social interactions or network size, and subjective social isolation, often referred to as loneliness, which reflects an individual’s internal perception of being emotionally disconnected from others, regardless of actual social contact [10,11]. While objective isolation is characterized by structural aspects (e.g., living alone, limited social contact), subjective isolation captures the unpleasant feelings of exclusion or emotional detachment.

In the context of urinary and fecal incontinence, as well as living with an ostomy, both forms of isolation are significant and can coexist. Individuals may avoid social situations due to embarrassment, fear of leakage or odor, and stigmatizing attitudes, leading to reduced social participation and concealment behaviors. Even when surrounded by others, they may experience profound loneliness, with feelings of being misunderstood or ashamed [12,13,14].

Social isolation carries significant public health implications. It is associated with increased risks of depression, cognitive decline, falls, and reduced quality of life. It also contributes to delayed help-seeking, diminished self-care, and greater risk of institutionalization [15,16]. As the World Health Organization emphasizes social participation as a key determinant of health, understanding and addressing the social isolation linked to incontinence is essential for improving both individual outcomes and population-level well-being [17].

Despite growing recognition of the psychosocial burden associated with urinary and fecal incontinence, as well as life with an ostomy, the literature remains fragmented and conceptually inconsistent. Definitions and measures of social isolation vary widely, often conflating subjective loneliness with objective social withdrawal. Moreover, few studies apply a theoretical framework to explain the link between incontinence and social isolation, limiting the comparability and clinical relevance of findings. Finally, many studies examine isolated aspects, such as psychological distress, stigma, or quality of life, without offering an integrated understanding of how these factors contribute to social isolation.

Thus, this scoping review aims to map the phenomenon of social isolation among individuals living with urinary and/or fecal incontinence, including those with an ostomy. This review is guided by the Middle Range Theory of Social Isolation in Chronic Illness [18], which provides a framework for understanding how chronic conditions contribute to or are influenced by social disconnection. The model differentiates between predisposing factors, reflecting stable sociodemographic and psychosocial vulnerabilities, and precipitating factors, referring to illness-related events that may trigger or worsen isolation. The outcomes encompass psychological, social, and health-related dimensions, capturing the wide-ranging effects of isolation on individual well-being. Guided by this theoretical framework, the present review was designed to explore how these constructs have been investigated in the context of incontinence and ostomy.

The research questions were the following: (i) How is social isolation and/or loneliness defined and operationalized in studies involving people with incontinence? (ii) What factors contribute to social isolation in this population? (iii) What are the consequences of social isolation and loneliness (e.g., mental health, quality of life, healthcare use)?

## 2. Materials and Methods

Scoping reviews are designed to explore and map key concepts and assess studies within a research domain, offering a comprehensive overview of the scope and characteristics of the existing literature [19]. This scoping review was conducted following the Joanna Briggs Institute (JBI) Manual for Evidence Synthesis [20] and the PRISMA-ScR checklist for its construction [21]. The research question was formulated using the Participant-Problem/Concept/Context (PCC) framework [20] which guided the development of the eligibility criteria, keywords, and search strategies (Table 1).

### 2.1. Operationalization of the Concepts

To guide the definition of social isolation and the classification of antecedents and outcomes, we referred to the Middle Range Theory of Social Isolation in Chronic Illness [18], which offers a context-specific lens to understand the construct and interpret the pathways through which chronic health conditions may lead to or be influenced by social isolation. Within this framework, antecedents are distinguished as predisposing factors, representing stable sociodemographic and psychosocial characteristics that shape baseline vulnerability, and precipitating factors, denoting illness-related conditions or events that can initiate or intensify social isolation. The outcomes of social isolation are conceptualized as psychological, social, and health-related consequences that together reflect the broad impact of isolation on individual well-being and clinical trajectories.

### 2.2. Eligibility Criteria

The inclusion criteria used for study selection were guided by the PCC framework [20] (Table 1). Eligible studies included primary research and review articles employing qualitative, quantitative, or mixed-methods designs that explored aspects of social isolation, loneliness, social participation, or social interaction in individuals living with an ostomy, urinary incontinence, or fecal incontinence. Studies were included if they involved adult participants (≥18 years), explicitly addressed social or psychosocial dimensions related to the target conditions, were published in English or Italian, and were peer-reviewed full-text articles.

Grey literature such as conference abstracts, dissertations, editorials, and commentaries was excluded due to the absence of peer review and the difficulty in assessing methodological quality. Studies focusing exclusively on clinical or surgical outcomes without reference to social or psychological aspects, as well as non-human studies, were also excluded.

#### Limits

In the database searches, a language filter was applied to include only studies published in English or Italian, while no time restriction was used to capture all available literature on the topic. In Cumulative Index to Nursing and Allied Health Literature (CINAHL) and PsychInfo databases the filters “exclude dissertations” and “exclude MEDLINE records” were applied to avoid grey literature and duplicate records already retrieved through the MEDLINE search.

### 2.3. Search Strategies

A three-step search strategy was implemented in accordance with JBI guidelines [20]. First, a preliminary search of MEDLINE (via PubMed) and CINAHL was performed to identify relevant keywords and indexed terms. Second, a comprehensive search using all identified terms was conducted across MEDLINE, CINAHL, and PsycINFO in July 2024 and updated in May 2025. Finally, the reference lists of the studies included were screened to identify additional records. Search strategies were developed in collaboration with a university research librarian and included both controlled vocabulary (e.g., MeSH terms) and free-text keywords, adapted to the syntax and indexing of each database. The main search concepts combined terms related to social connectedness (“social isolation,” “loneliness,” “social participation,” “social interaction”) and stoma- or incontinence-related conditions (“ostomy,” “surgical stoma,” “urinary incontinence,” “fecal incontinence”). Boolean operators (AND/OR) were used to combine these concepts appropriately, ensuring sensitivity and specificity across databases (Appendix A).

### 2.4. Document Selection

The study selection process followed the PRISMA 2020 flow diagram [22] (Figure 1). Screening was conducted using the web application Rayyan (MA, USA) [23] in two steps. The first step involved selecting records based on title and abstract by two independent reviewers (V.S., I.M.). After this preliminary selection, full texts of the selected records were assessed independently by two reviewers (V.S., I.M.). Any discrepancies were resolved through discussion or with the assistance of a third senior reviewer (G.V.). Records were systematically collected, duplicates removed using Zotero (version 7.0.27; VA, USA) [24], and manually sorted. In line with JBI guidance, no formal quality appraisal was performed [25].

### 2.5. Data Extraction

Metadata from the selected records were extracted using Zotero reference manager software (version 7.0.24) [24] and manually verified upon import. A data extraction table was developed based on the JBI guidelines [20] and subsequently expanded to include additional variables of interest. Specifically, data extraction captured the presence of theoretical models adopted to conceptualize social isolation, as well as the identification of its antecedents and outcomes. All data were manually extracted and organized in a Microsoft Excel spreadsheet (version 16.98).

### 2.6. Data Presentation

Data were analyzed thematically to identify recurring concepts and patterns. The classification of antecedents and outcomes of social isolation was organized and presented according to the Middle Range Theory of Social Isolation in Chronic Illness [18].

## 3. Results

### 3.1. Selection Process Description

The database searches yielded a total of 314 records. After removing 26 duplicates and 36 records excluded for other reasons (e.g., non-research material or unavailable language), 252 records remained for screening. During title and abstract screening, 149 records were excluded as not pertinent to the research question.

A total of 103 potentially eligible records were sought for full-text retrieval. Four of these could not be accessed despite assistance from the institutional library service and attempts to contact the corresponding authors. Consequently, 99 full-text articles were assessed for eligibility. Of these, 80 were excluded for not meeting the predefined inclusion criteria, specifically, 43 for not addressing the target concept, 9 for non-eligible population, and 28 for aims unrelated to the review question. Ultimately, 23 studies met all criteria and were included in the final synthesis. The selection process is summarized in Figure 1.

### 3.2. Characteristics of the Included Studies

Table 2 summarizes the studies included in the review, highlighting their design, setting, aims, sample characteristics, and main findings related to social isolation.

The included articles were published between 1984 and 2024, with a publication peak between 2015 and 2022. Of the 23 records, 12 were quantitative studies [34,35,36,37,38,39,40,41,42,43,44,45], eight (34.78%) were qualitative studies [26,27,28,29,30,31,32,33,49], and three (13.04%) were reviews (two metasyntheses and one systematic review) [46,47,48]. Sample sizes varied according to study design: qualitative studies included 11–33 participants, whereas quantitative studies enrolled 303–16,369 participants. The reviews synthesized between 20 and 95 articles. Participants ranged in age from 6 to 97 years, with most samples predominantly composed of women (female representation: 51.5–100%). Only one study focused on a pediatric population [32].

Regarding health conditions, 13 studies (56.5%) addressed urinary incontinence, one (4.3%) focused on fecal incontinence, four (17.4%) examined individuals with an ostomy, and five (21.7%) investigated combinations of these conditions. Social isolation was the primary outcome in 14 studies (61%), while in the remaining nine (39%) it emerged as a secondary theme (Appendix A).

### 3.3. Results Synthesis

Driven by the Middle Range Theory of Social Isolation [18], the findings were organized into three subthemes: conceptualization of social isolation, antecedents of social isolation, and outcomes of social isolation.

#### 3.3.1. Conceptualization of Social Isolation

None of the included studies explicitly adopted a theoretical framework to define social isolation. Nevertheless, three recurrent dimensions emerged across the evidence base: (i) subjective isolation, characterized by individual perceptions of emotional or relational disconnection from others [27,29,34,40,41,42,43,45]; (ii) interaction-related isolation, described as discomfort in social interactions, avoidance of participation, and progressive withdrawal from interpersonal relationships [26,28,30,31,35,37,39,44,47,48,49]; and (iii) objective or structural isolation, associated with tangible barriers such as physical, environmental, or logistical constraints [33,36,38,46]. In this dimension, social isolation was frequently operationalized through behavioral indicators such as being homebound (going out once per week or less), withdrawal from structured activities, or reduced participation in regular social interactions [29,36]. One study [32] also suggested that social isolation may extend to parents and caregivers, who can experience emotional strain and withdrawal related to the burden of care.

Quantitative studies most often measured isolation using the UCLA Loneliness Scale [34,41] and the De Jong Gierveld Loneliness Scale—Short Form [45]. The UCLA scale assesses perceived isolation, lack of companionship, and social exclusion, whereas the De Jong Gierveld scale captures both emotional and social loneliness, which include the absence of close emotional bonds and perceptions of insufficient contact with other community members.

#### 3.3.2. Antecedents of Social Isolation

##### Predisposing Factors

Social participation emerged in several studies as a key protective factor against isolation. Takahashi et al. [29] described motivation to engage with others as a resource that fosters activation and relational resilience, even in the presence of physical or psychosocial limitations. However, other studies have emphasized that when social exchanges are perceived as superficial or lacking genuine interest, they can paradoxically reinforce feelings of distance and alienation [27,48].

Psychological resilience was also highlighted as relevant in the context of incontinence. Takahashi et al. [29] underscored the importance of the Sense of Coherence (SOC), particularly the “meaningfulness” component, in helping individuals maintain a sense of purpose and resist disengagement. These findings suggest that resilience resources may mitigate, though not completely eliminate, experiences of isolation.

In addition, social attitudes contributed to variations in isolation. Fultz and Herzog [40] reported that individuals with a stronger desire for social approval were less likely to acknowledge the emotional impact of urinary incontinence on their self-perception. This suggests that some individuals may downplay or minimize their difficulties to conform to social expectations and avoid stigmatization.

Environmental accessibility of toilet facilities was a recurring factor in determining isolation. A lack of private, clean, and accessible toilets was repeatedly associated with greater social withdrawal [29,33]. Many individuals reported structuring their daily routines around toilet availability and avoiding public or social settings for fear of not being able to manage incontinence episodes. This dependency, described as a “toilet-bound” condition, restricted social mobility and contributed to disengagement [27]. Conversely, the presence of secure and accessible toilets promoted confidence and participation [38]. Similar dynamics were observed in schools and workplaces, where inadequate facilities limited educational involvement, peer interactions, and professional activities, sometimes resulting in reduced working hours, role changes, or job loss [26,27,33,49]. Consistently, the systematic review by Capilla-Díaz et al. [46], emphasized that individuals with intestinal ostomy face persistent challenges related to odor control and job security.

Socioeconomic characteristics, such as education and income, also emerged as structural determinants of isolation. Lower educational attainment was significantly associated with greater loneliness among individuals with incontinence [40], suggesting that education may act as a protective resource. Similarly, low income was linked to higher isolation levels among people with an ostomy [37].

Gender-related findings were mixed. Some studies reported no differences in perceived loneliness between men and women [41,45], whereas others found that men were more likely to experience activity limitations due to urinary incontinence [40]. Conversely, Li et al., [37] reported a higher incidence of social isolation among women with permanent colostomies [37].

Finally, age was consistently associated with social participation. Older adults were more likely to experience reduced social participation, partly due to functional limitations, comorbidities, and increased reliance on environmental supports [38].

##### Precipitating Factors

Non-acceptance of one’s condition emerged as a key determinant of social isolation. Among individuals with an ostomy, difficulties in adapting to the device were frequently linked to loneliness, loss of family roles, and reduced intimacy within couples, often resulting in estrangement and social withdrawal [27,46].

Illness trajectory also influenced experiences of isolation. Participants in Pape et al. [27] described loneliness as most pronounced in the period immediately following stoma creation and cancer treatment, with gradual improvement over time as coping strategies and hope developed—though isolation never fully disappeared. Similarly, among incontinent cancer patients, loneliness peaked during the early post-treatment phase, gradually lessening with adaptation but often persisting over time [27,39,46].

A diminished sense of belonging was another recurrent theme. In Malawi, women with obstetric fistula (a childbirth-related injury leading to continuous urinary or fecal incontinence) reported profound exclusion from family and community life, which severely undermined both their social identity and personal sense of belonging [28]. Likewise, patients with colorectal cancer described how social isolation weakened interpersonal ties and eroded feelings of belonging in everyday relationships [37].

In older adults with incontinence, low social support more than doubled the likelihood of experiencing loneliness [41]. Conversely, in individuals with Inflammatory Bowel Disease (IBD), informal networks of family, friends, or peers were reported to reduce loneliness and improve adjustment [48]. Interestingly, Takahashi et al. [38] observed that greater perceived support was sometimes associated with being homebound, possibly reflecting that those with greater functional limitations and existing social isolation are also those who receive more assistance, rather than suggesting that support itself leads to isolation.

Stigma consistently emerged as one of the most prominent precipitating factors. Shame and fear of judgment discouraged disclosure, encouraged avoidance, and reinforced isolation [27,28]. Anticipated stigma often led individuals to limit activities to avoid embarrassment. For example, MacDonald & Anderson [39] reported that many patients had never disclosed the presence of an ostomy even to close family members or partners. Similar concealment behaviors were observed among women with obstetric fistula, who deliberately restricted social interactions to minimize the risk of symptom exposure [28]. Children with spina bifida also reported avoiding even trusted social contexts for fear that peers might discover their condition [32]. Among older adults, many deliberately withdrew from social life to conceal incontinence and prevent gossip or public accidents [31]. Comparable patterns were observed among colostomy patients, where fear of odors, noises, or stigmatization led to reduced social and professional engagement, reluctance to disclose their condition, and even avoidance of intimacy [46]. These concealment strategies often lead to voluntary self-exclusion, which is used as a means of protection against rejection or humiliation.

Self-esteem was frequently reported as a negative determinant of isolation. In the Malawian study, women described developing deeply negative self-perceptions—either after others reacted to accidental symptom disclosure or after witnessing the public humiliation of peers with similar conditions. These experiences fostered feelings of inferiority, eroded self-confidence, and prompted further withdrawal, with some participants even describing themselves as “not real persons” or “damaged,” equating bodily dysfunction with a diminished social identity and human worth [28]. Among older adults with incontinence, poor symptom management was linked to reduced self-esteem and greater social withdrawal, whereas effective control was associated with improved participation and reduced isolation [31]. Likewise, in rectal cancer patients, social isolation was directly associated with lower self-esteem [39].

Clinical severity and illness-related conditions were consistently associated with greater social isolation. Individuals with more severe incontinence were more likely to experience significant social restrictions and loneliness compared to those with milder symptoms [40]. Several studies [34,44] suggested that disease severity, together with illness-related stigma, amplified the psychological burden of incontinence and intensified feelings of isolation. Qualitative evidence confirmed that severe signs and symptoms, such as visible leakage, noticeable odors, and urgency, limited social participation and encouraged withdrawal [33,46]. Interestingly, in some cases, the severity of symptoms served as a trigger for help-seeking, prompting individuals to seek medical or professional support [47].

Reduced autonomy also emerged as an important factor. Takahashi et al. [38] reported that individuals with urinary incontinence who were dependent on walking aids were more likely to be homebound, indicating that mobility impairment is a substantial risk factor for social isolation. Similarly, reduced independence in managing incontinence was linked to greater withdrawal, whereas individuals able to self-manage their symptoms experienced better social integration [29,31,49]. Those reliant on caregivers were reported to be excluded more frequently, suggesting that autonomy plays a protective role against isolation.

##### Outcomes of Social Isolation

The outcomes of social isolation were consistently negative, spanning psychological, social, and quality-of-life domains.

Psychosocial consequences were the most frequently reported, with isolation associated with sadness, depression, and anxiety [27,28,40,45]. Participants often described loss of motivation, emotional estrangement from others, and erosion of intimacy within couples [27,28]. Internalized shame deepened these effects, leading to guilt, a fractured sense of identity, and feelings of being “damaged” or “abnormal” [28,32,39,48]. Concealment and avoidance strategies reinforced this emotional distance, often extending even to close relationships [39].

Quality of life was consistently reported as poor among individuals experiencing social isolation [27,44,46]. Participants frequently described feeling confined to their homes, with the unpredictability of symptoms—especially bowel urgency—restricting daily life and reinforcing a sense of entrapment [27]. Quantitative analyses confirmed that isolation was a major contributor to reduced quality of life [44].

Overall, the evidence indicates that social isolation in people with incontinence is not a peripheral issue but a core component of their lived experience, one that profoundly shapes mental health and overall well-being. Figure 2 summarises our findings.

## 4. Discussion

This scoping review systematically mapped the literature on social isolation among individuals living with urinary or fecal incontinence and those with an ostomy, using the Middle Range Theory of Social Isolation in Chronic Illness [18] as an interpretative framework. To our knowledge, this is the first review to provide a structured synthesis of the antecedents and outcomes of social isolation in this population, offering both clinical and theoretical insights.

From a theoretical standpoint, our findings reveal that none of the included studies explicitly applied a conceptual framework to define or analyze social isolation. This lack of theory-driven approaches limits comparability and interpretability across studies. By applying the Middle Range Theory of Social Isolation, this review demonstrates its usefulness in organizing heterogeneous evidence and highlights the need for future research to systematically consider both subjective and objective dimensions of isolation.

The Middle Range Theory of Social Isolation [18] conceptualizes social isolation as a multidimensional phenomenon encompassing both loneliness (as subjective emotional experience) and social disconnectedness (an objective reduction in social contact and participation). Our synthesis indicates that, in the context of incontinence, research has focused predominantly on the subjective dimension—loneliness—while objective indicators, such as reduced participation in activities or homebound status, have been less frequently investigated. This gap highlights the need for future studies to systematically address both dimensions to provide a more comprehensive understanding of isolation in this population.

The findings by Stickely [34] show that loneliness can occur even in the presence of an active social network when there is a mismatch between desired and actual relationships. For individuals with incontinence, this mismatch often reflects compromised parental or spousal roles, disrupted intimacy, and weakened family belonging. When communication, sexuality, or emotional reciprocity are impaired, relationships may lose emotional value, fostering feelings of inadequacy, detachment, and estrangement [27]. The feeling of being misunderstood—especially regarding invisible or trivialized symptoms—can further intensify isolation, even in the context of close relationships [29,48]. Interactions perceived as insensitive or humiliating may be experienced as discouraging, prompting emotional withdrawal and progressive disengagement, which over time consolidate a state of social disconnection [27].

Moreover, it is important to acknowledge that social isolation does not affect only individuals living with incontinence or those with an ostomy but also extends to their close relatives and caregivers. Fischer’s study [32] suggests that social isolation may also affect parents and caregivers of incontinent patients, who can experience emotional strain and social withdrawal related to the burden of care. This finding points to the relational nature of isolation and highlights the need for further research on caregivers’ experiences.

Consistent with previous literature on social isolation in other chronic conditions [50,51], our synthesis indicates that predisposing factors represent structural vulnerabilities that reduce individuals’ capacity to cope with the challenges of incontinence and increase their risk of isolation. Among these, toilet accessibility and socioeconomic resources were the most frequently reported. Toilet accessibility emerged as a particularly significant determinant in this population [32,46]. Qualitative studies described the state of “being toilet-bound” not only as a practical limitation but also as a symbolic marker of fragility and uncertainty, which undermines a person’s sense of control and social mobility [27]. Consequently, many individuals avoid going out, not because it is impossible, but because they feel unable to do so while maintaining dignity. Similarly, lower education and limited income were repeatedly associated with greater isolation [13,40]. Financial constraints can severely restrict access to continence products, medical care, and supportive environments, thereby limiting the ability to manage symptoms discreetly and increasing the likelihood of withdrawal and social disengagement [52].

Among precipitating factors, stigma consistently emerged as the most prominent mechanism. In the broader literature on chronic illness, stigma has been widely examined, particularly in conditions that are not outwardly visible [53]. Evidence from diseases such as psoriasis and cancer shows that stigma—whether internalized, anticipated, or enacted—can strongly mediate quality-of-life outcomes [54,55]. Similar dynamics were observed among individuals with incontinence, where the concealable yet socially stigmatized nature of the condition fosters shame, concealment, and identity fracture [27,28]. Anticipated stigma frequently led to self-exclusion as a strategy to avoid humiliation [39].

Unlike conditions such as psoriasis or cancer, which may trigger fears of contagion or death and lead to overt social rejection, incontinence poses no risk to others—yet still provokes withdrawal and self-devaluation. Cultural context appears to influence these dynamics: for instance, the Malawian study [28] demonstrated how social norms related to marriage, fertility, and gender roles intensified fears of rejection and abandonment, leading to profound isolation. In Western contexts, stigma was more commonly described in relation to practical barriers or psychological responses. These findings support the idea that stigma stems less from the inherent danger of the condition and more from its visibility and the way it conflicts with cultural expectations around bodily control and social decorum [44].

The studies included in this review reinforce findings from the broader literature on chronic conditions [51]: social isolation is not merely an emotional experience but also exacerbates the negative effects of illness on mental health [27,28,40,45]. Furthermore, evidence indicates that isolation significantly compromises quality of life among individuals with incontinence [27,44,46], mirroring patterns observed in other chronic conditions [56]. These findings underscore that well-being depends not only on physical health but also on the ability to maintain meaningful relationships. Interventions aimed at reducing predisposing and precipitating factors for isolation may therefore contribute to improving quality of life.

### 4.1. Implications for Practice and Research

This review highlights that social isolation among individuals living with incontinence is a complex, multidimensional phenomenon with significant clinical and psychosocial implications. Identifying the factors that contribute to isolation and understanding their consequences provide an opportunity to develop targeted interventions. Screening for social isolation using validated tools could be incorporated into routine continence care pathways, enabling timely identification of at-risk individuals. Specialized nurses, such as stomal therapy nurses, play a key role not only in providing clinical management but also in supporting patients’ reintegration into social life. At the policy level, these practices should be aligned with public health strategies: both the World Health Organization (WHO) [57] and the U.S. National Academies of Sciences, Engineering, and Medicine (NASEM) [50] recognize social isolation as a social determinant of health, associated with higher morbidity, premature mortality, and increased healthcare utilization. Incorporating screening and supportive practices into clinical pathways, therefore, contributes to broader goals of reducing health inequalities, improving quality of life, and promoting population well-being.

From a research perspective, this review supports the use of the Middle Range Theory of Social Isolation in Chronic Illness as a conceptual framework to interpret the experiences of individuals with incontinence. This theory can guide future hypothesis-driven investigations by clarifying the mechanisms through which predisposing and precipitating factors lead to isolation and by identifying mediating processes such as self-concept erosion and loss of perceived control. However, important gaps remain. Objective dimensions of isolation, such as participation frequency and network size, are underexplored and should be systematically investigated. Moreover, health-related behaviors (e.g., self-care) and clinical outcomes of isolation remain insufficiently studied. Future research should include longitudinal designs to track the development and persistence of isolation over time, as well as interventional studies to test effective strategies for its prevention and reduction. Promising approaches may include structured peer-support groups, digital tools to foster connection, and psychoeducational interventions to empower patients and caregivers in coping with the relational and emotional consequences of incontinence.

### 4.2. Limitations

This review has several limitations. First, only studies published in English or Italian were included, which may have led to the exclusion of relevant evidence published in other languages. This restriction was applied because these are the languages spoken by all members of the research team, allowing accurate interpretation of the data without the risk of translation errors. Second, although both qualitative and quantitative studies were considered, their heterogeneity in terms of population characteristics, cultural context, type of incontinence, and study design limited direct comparability of results. Nevertheless, this variability also enriched the analysis by providing a more comprehensive picture of the phenomenon across diverse settings and populations. Third, the heterogeneity of the included populations represents a key limitation of this review. Most studies focused on urinary incontinence, while ostomy-related evidence remains scarce and largely qualitative. Furthermore, none of the studies distinguished between urinary incontinence subtypes (e.g., overactive bladder vs. stress urinary incontinence), preventing more detailed comparisons. This imbalance restricts generalizability and precludes analysis of condition-specific mechanisms. Future research should examine these subgroups separately to capture differential psychosocial pathways and tailor interventions accordingly. Finally, the review protocol was not pre-registered on a public repository. While registration is not mandatory for scoping reviews, it is increasingly recognized as good practice to enhance methodological transparency and reproducibility.

## 5. Conclusions

This scoping review provides the first systematic synthesis of evidence on social isolation in individuals with urinary or fecal incontinence and those living with an ostomy. The findings demonstrate that isolation is not a marginal or secondary experience, but a core dimension of the lived experience of incontinence, closely linked with stigma, shame, and withdrawal and associated with adverse mental health outcomes and reduced quality of life. By applying the Middle Range Theory of Social Isolation in Chronic Illness, this review highlights the value of theory-informed approaches for integrating fragmented evidence and guiding both research and clinical practice. Social isolation should be systematically assessed and addressed within continence care pathways as part of person-centered, holistic care.

Future research should adopt longitudinal and interventional designs to clarify causal mechanisms, explore objective indicators of isolation, and evaluate strategies to reduce its impact. Interventions, whether educational, clinical, or community-based, should aim not only to improve continence management but also to promote social participation and protect mental health.

## Figures and Tables

**Figure 1 nursrep-15-00375-f001:**
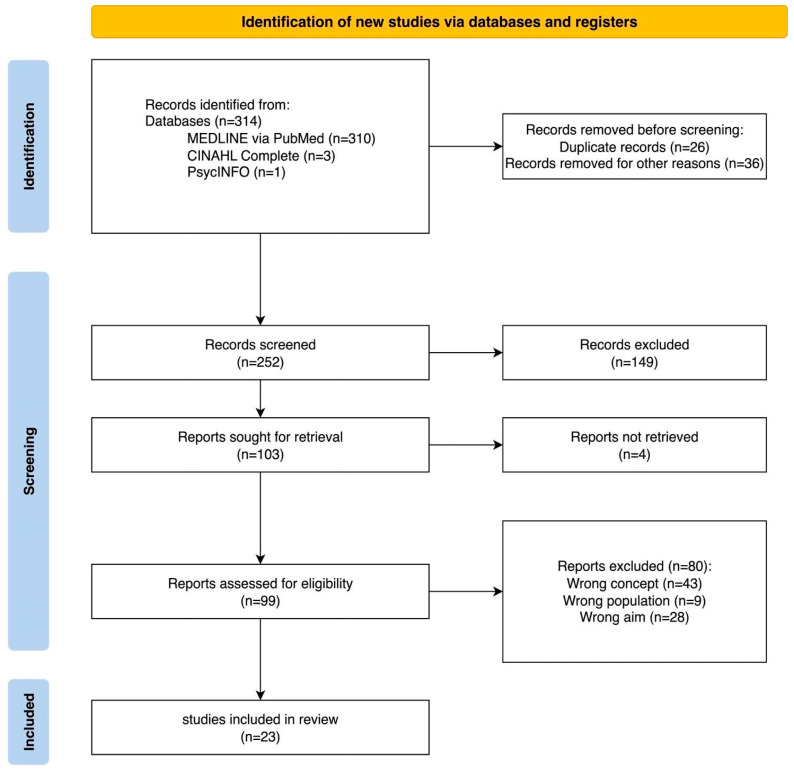
PRISMA Flow Diagram 2020: selection process.

**Figure 2 nursrep-15-00375-f002:**
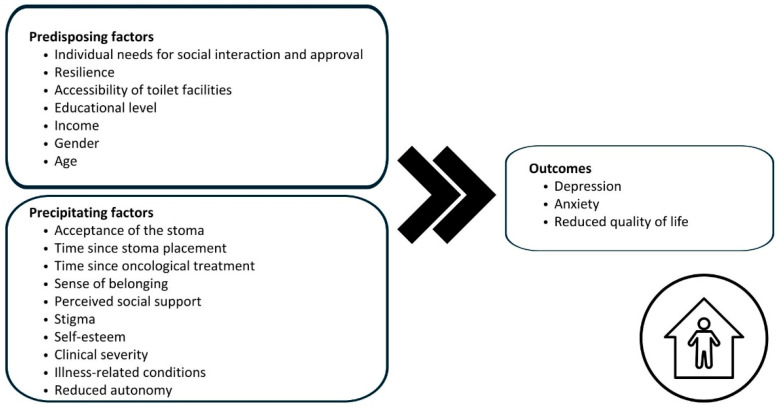
Antecedents and outcomes of social isolation in individuals living with incontinence.

**Table 1 nursrep-15-00375-t001:** PCC Framework.

PCC	
Population	Individuals living with urinary and/or fecal incontinence and people living with urinary and/or intestinal ostomy
Concept	Social isolation
Context	Hospital, homes and community settings

**Table 2 nursrep-15-00375-t002:** Summary of the included studies.

Author(s) (Year)	Title	Country	Design	Context/Aim	Sample	Lived Experiences Related to Social Isolation
Qualitative study						
Esparza et al. (2018) [26]	Experiences of women and men living with urinary incontinence: A phenomenological study	Spain	Interpretative Phenomenological Analysis (IPA)	Urodynamics Unit, Murcia. Aim: To explore gender-based quality-of-life experiences of people with UI.	33 (17 women, 16 men)	Social isolation was a key theme, linked to giving up normal activities and concealing the condition out of embarrassment and fear. Men feared ridicule, especially at work; avoiding disclosure prevented stigma.
Pape et al. (2022) [27]	The trajectory of hope and loneliness in rectal cancer survivors with major low anterior resection syndrome	Belgium	Grounded Theory	Three teaching hospitals. Aim: To explore hope and loneliness trajectories in LARS survivors.	28 (18 men, 10 women)	Loneliness emerged at physical, family, and social levels: being toilet-bound, restricted family life, and avoidance of public activities. Shame and fear of leakage led to withdrawal and isolation.
Changole et al. (2017) [28]	“I am a person but I am not a person”: Experiences of women living with obstetric fistula in Malawi	Malawi	Qualitative Study	Bwaila Fistula Care Center. Aim: To explore stigma and lived experiences of women with obstetric fistula.	25 women + 20 relatives	Participants faced anticipated, internalized, enacted, and associative stigma, leading to shame, concealment, low self-esteem, and withdrawal. Anticipated stigma was universal; some women faced abandonment.
Takahashi et al. (2016) [29]	Psychological resilience and active social participation among older adults with incontinence	Japan	Qualitative Study	Community-based study in Chiba. Aim: To describe resilience and social participation in older adults with incontinence.	11 adults	Motivation for social activity was hindered by psychological stress but supported by desire to interact, exercise, and confidence in managing symptoms. Accessibility reduced fear of going out and isolation risk.
Komorowski et al. (2007) [30]	Female Urinary Incontinence in China: Experiences and Perspectives	China	Interpretative Phenomenological Analysis (IPA)	Hospital outpatient clinic, Fujian. Aim: To understand experiences and perspectives of Chinese women with UI.	15 women	Emotional isolation and avoidance were common. Shame, sadness, and fear of public exposure led to staying home and reducing social or physical activities, reinforcing isolation.
Mitteness (1987) [31]	The Management of Urinary Incontinence by Community-Living Elderly	USA	Ethnographic Study	Urban senior housing. Aim: To examine beliefs, management, and social responses to UI among older adults.	30 adults	Most managed incontinence through isolation to preserve dignity. Active individuals followed strict routines; frailer ones became housebound. Home privacy redefined “control,” reinforcing isolation.
Fischer et al. (2015) [32]	Experiences of children with spina bifida and their parents around incontinence and social participation	Canada	Qualitative Interview Study	Pediatric rehabilitation hospital. Aim: To explore continence, participation, and peer relationships in spina bifida.	11 children + 10 parents	Social isolation was common among both children and parents. Concealment and stigma limited peer interaction. Independence promoted inclusion, while dependence increased exclusion.
Johnsen et al. (2009) [33]	Problematic aspects of faecal incontinence according to the experience of adults with spina bifida	Norway	Qualitative Interview Study	National Resource Centre for Spina Bifida. Aim: To explore the social consequences of faecal incontinence.	11 adults	Incontinence restricted education, social, and intimate life. Fear of leakage and odor caused shame and withdrawal. Despite family support, fear of rejection reinforced isolation.
**Author(s) (Year)**	**Title**	**Design**	**Country**	**Aim**	**Sample**	**Main Findings on Social Isolation**
Quantitative Studies						
Stickley et al. (2017) [34]	Urinary Incontinence, Mental Health, and Loneliness Among Older Adults	Cross-sectional	Ireland	To examine the association between urinary incontinence (UI), loneliness, and mental health	6903 adults	UI was significantly associated with higher loneliness (unadjusted OR = 1.74, *p* < 0.001; adjusted OR = 1.51, *p* < 0.001). After adding depression, the effect became non-significant (OR = 1.20, *p* > 0.05), indicating that depression mediates the relationship between UI and loneliness. Activity limitations due to UI remained significantly related to loneliness until depression was included in the model (*p* < 0.01).
Wang et al. (2015) [35]	Effects of stigma on Chinese women’s attitudes toward seeking treatment for urinary incontinence	Cross-sectional (community-based)	China	To examine how stigma influences attitudes toward treatment and whether effects vary by symptom severity	305 women	Social rejection was positively associated with social isolation (*p* < 0.01), and higher isolation was associated with more negative treatment attitudes (*p* < 0.01). The mediation model explained 22% of the variance in social isolation and 28% in treatment attitudes.
Park et al. (2022) [36]	Urinary Incontinence and Depressive Symptoms: The Mediating Role of Physical Activity and Social Engagement	Cross-sectional secondary analysis	South Korea	To examine the mediating role of physical activity and social engagement between UI and depressive symptoms	1327 women	UI was associated with fewer social connections (β = −0.825, 95% CI [−1.609, −0.041], *p* < 0.05) and lower participation in social activities (β = −0.080, *p* < 0.05). Social engagement mediated approximately 20% of the association between UI and depressive symptoms; combined with physical activity, the total mediated effect reached 22% (*p* < 0.01).
Li et al. (2024) [37]	Linking stigma to social isolation among colorectal cancer survivors with permanent stomas	Cross-sectional	China	To examine the mediating role of stoma acceptance and valuable actions in the association between stigma and social isolation	303 adults	Participants reported moderate-to-high levels of social isolation (mean = 14.91 ± 3.83). Perceived stigma was positively correlated with social isolation (r = 0.30, *p* < 0.001). Mediation analysis showed two indirect pathways: (1) stigma increased isolation indirectly by reducing engagement in valuable social actions (β = 0.057, 95% CI [0.018–0.107], *p* < 0.01; 21% of total effect), and (2) a sequential pathway where stigma lowered stoma acceptance, which in turn reduced valuable actions, leading to greater isolation (β = 0.036, 95% CI [0.014–0.062], *p* < 0.01; 13% of total effect).
Takahashi et al. (2015) [38]	Sense of Coherence as a Key to Improve Homebound Status Among Older Adults with Urinary Incontinence	Cross-sectional	Japan	To assess the association between sense of coherence and being homebound among older adults with UI	411 adults	Among participants with UI, 32.9% were homebound. Higher levels of meaningfulness (a subdimension of sense of coherence) were associated with lower odds of being homebound (OR = 0.79, 95% CI [0.65–0.96], *p* < 0.05). Walking dependence increased the risk (AOR = 3.77, *p* < 0.01), and higher perceived social support was also associated with homebound status (AOR = 1.07, *p* < 0.05).
MacDonald et al. (1984) [39]	Stigma in Patients with Rectal Cancer: A Community Study	Cross-sectional community survey	UK	To explore perceived stigma among rectal cancer patients and its psychosocial implications	420 adults	Patients with colostomy reported higher stigma (*p* < 0.001) than those with anastomosis. Self-consciousness (40% vs. 17%, *p* < 0.01) and feeling different (14% vs. 5%, *p* < 0.05) were more frequent. Stigma was associated with greater depression (38% vs. 12%) and anxiety (42% vs. 10%), reduced social activity (73% vs. 51%), and poorer marital and sexual relations (*p* < 0.001).
Fultz et al. (2001) [40]	Self-Reported Social and Emotional Impact of Urinary Incontinence	Cross-sectional	USA	To assess social and emotional correlates of UI	1322 adults	UI was significantly associated with loneliness (OR = 2.10, 95% CI [1.38–3.16], *p* < 0.001). Greater urine loss predicted more social restriction (OR = 3.47, *p* < 0.01). Loneliness was more frequent among those with lower education (*p* < 0.05) and poorer health.
Ramage-Morin et al. (2013) [41]	Urinary Incontinence and Loneliness in Canadian Seniors	Cross-sectional	Canada	To estimate prevalence of UI and its association with loneliness	16,369 adults	Adults with UI had higher odds of loneliness (OR = 1.8, 95% CI [1.5–2.0], *p* < 0.001; fully adjusted AOR = 1.5, 95% CI [1.3–1.7]). The association remained significant after adjusting for social support and disability and did not differ by sex.
Yip et al. (2013) [42]	The Association Between Urinary and Fecal Incontinence and Social Isolation in Older Women	Cross-sectional secondary analysis	USA	To investigate the relationship between UI, FI, and social isolation	1412 women	Women with daily UI were more likely to report isolation (6.6% vs. 2.6%; adjusted OR = 3.0, *p* = 0.03) and had higher UCLA loneliness scores (*p* = 0.003). Weekly FI was initially associated with isolation (11.6% vs. 2.8%, *p* = 0.01) but became non-significant after adjustment (*p* = 0.65).
Nichols (2011) [43]	Social Connectivity in Those ≤24 Months Post-Surgery	Cross-sectional survey	North America and UK	To evaluate social isolation, emotional support, and life satisfaction in adults post-ostomy	560 adults	About 20% were socially isolated, mainly in the early postoperative months. Isolation decreased significantly over 24 months (*p* = 0.020). Emotional support was strongly inversely correlated with isolation (r = 0.65, *p* < 0.001), and positive body image and nurse support were associated with better social connectivity.
Wan et al. (2014) [44]	Disease Stigma and Its Mediating Effect on Symptom Severity and Quality of Life in Stress Urinary Incontinence	Cross-sectional descriptive	China	To test whether disease stigma mediates the relationship between symptom severity and quality of life	333 women	Social isolation scores were moderate (mean = 2.23 ± 0.51) and positively correlated with symptom severity, while negatively correlated with quality of life (r = −0.45, *p* < 0.001). Social isolation and internalized shame partially mediated the relationship between symptom severity and quality of life, reducing the direct effect by 34% (*p* < 0.001).
Zhang et al. (2019) [45]	Incontinence and Loneliness Among Older Adults with Multimorbidity	Cross-sectional	Hong Kong	To explore associations between incontinence and loneliness dimensions	741 adults	Incontinence was significantly associated with higher emotional loneliness (β = 0.35, 95% CI [0.07–0.64], *p* = 0.01), but not with social loneliness or total loneliness after adjustment for confounders. No gender interaction effects were observed.
**Author(s) (Year)**	**Title**		**Design**	**Aim**	**Sample**	**Main Results Related to Social Isolation**
Reviews						
Capilla-Díaz et al. (2019) [46]	Living With an Intestinal Stoma: A Qualitative Systematic Review	Qualitative systematic review		To explore experiences and coping processes among people living with intestinal stomas by synthesizing qualitative evidence	95 studies	Social isolation was commonly linked to embarrassment, altered body image, and fear of leakage or odor. Stoma-related stigma and loss of control led to avoidance of social events and public places, reducing participation and connectedness.
Yan et al. (2022) [47]	Perceptions and Help-Seeking Behaviours Among Community-Dwelling Older People With Urinary Incontinence: A Systematic Integrative Review	Systematic integrative review		To synthesize evidence on perceptions and help-seeking behaviours related to urinary incontinence among community-dwelling older adults (based on the COM-B model)	20 studies	Self-stigma and embarrassment were prevalent and associated with concealment of symptoms (*p* < 0.05). Stigma discouraged disclosure and treatment-seeking, reinforcing self-management and withdrawal. Social isolation increased among those lacking peer connection or support, reducing motivation to seek care (*p* < 0.05).
Kemp et al. (2012) [48]	Understanding the Health and Social Care Needs of People Living With IBD: A Meta-Synthesis of the Evidence	Meta-synthesis of qualitative studies		To synthesize qualitative research on the health and social care needs of individuals living with inflammatory bowel disease (IBD)	7 studies	Fear of incontinence and unpredictability of symptoms led to avoidance of social contexts. Restrictive behaviors (staying near toilets, avoiding public activities) resulted in social withdrawal, emotional distress, and reduced self-confidence. Isolation was accompanied by feelings of inadequacy, guilt, and strained family relationships.

Legend: IPA = Interpretative Phenomenological Analysis; UI = Urinary Incontinence; FI = Fecal Incontinence; LARS = Low Anterior Resection Syndrome; IBD = Inflammatory Bowel Disease; UCLA = University of California, Los Angeles Loneliness Scale; OR = Odds Ratio; AOR = Adjusted Odds Ratio; CI = Confidence Interval; p = p-value; β = Beta Coefficient; r = Correlation Coefficient; SD = Standard Deviation; n = Sample Size.

## Data Availability

No new data were created or analyzed in this study. Data sharing is not applicable to this article.

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
