# Peer review of "Social Isolation Among Individuals with Incontinence: A Scoping Review"

_nursrep, 2025, doi:10.3390/nursrep15110375_

Round 1

Reviewer 1 Report

Comments and Suggestions for Authors

Review of the manuscript "Social Isolation Among Individuals with Incontinence: A Scoping Review" submitted to Nursing Reports.

The manuscript covers a relevant and insufficiently researched topic, such as social isolation in people with urinary and faecal incontinence, as well as in people with a stoma. The idea itself, along with the research questions posed, has significant clinical and social potential. The authors opted for a scoping review, which is a methodologically appropriate choice for mapping phenomena and identifying existing research. The text is well-structured, the results are clearly presented, and the discussion is supported by a theoretical framework with the Middle Range Theory of Social Isolation in Chronic Illness, which can be considered a significant contribution.

However, I find several methodological and reporting weaknesses:

Namely, the manuscript does not provide sufficient detail on the search strategy. Merely listing the databases is insufficient; the exact keywords, Boolean operators, and search strings for each database must be explicitly reported. Therefore, I strongly recommend adding the full search strategy, either in the main text or in a supplementary table.

Additionally, there is no mention of a pre-registered protocol (e.g., OSF). While not mandatory for scoping reviews, protocol registration is increasingly recognised as best practice. If the protocol was not registered, this should be clearly acknowledged as a limitation.

Furthermore, the authors state that they followed the PRISMA-ScR guidelines, but several checklist elements are missing or underdeveloped. For example, the inclusion and exclusion criteria are not fully detailed, and the reasons for exclusion at each stage of the screening process are not provided. These details should be systematically reported to align with PRISMA-ScR standards.

Finally, I strongly recommend including a comprehensive table summarising the characteristics of the included studies (author, year, country, design, sample size, instruments used, and key findings). This would greatly facilitate visibility and allow readers to navigate more easily, which is a common practice in scoping review manuscripts.

In its current form, the manuscript is interesting and rich in content, but it requires additions and refinements to meet the standards for a fully comprehensive scoping review.

Author Response

The manuscript covers a relevant and insufficiently researched topic, such as social isolation in people with urinary and faecal incontinence, as well as in people with a stoma. The idea itself, along with the research questions posed, has significant clinical and social potential. The authors opted for a scoping review, which is a methodologically appropriate choice for mapping phenomena and identifying existing research. The text is well-structured, the results are clearly presented, and the discussion is supported by a theoretical framework with the Middle Range Theory of Social Isolation in Chronic Illness, which can be considered a significant contribution.

Thank you

However, I find several methodological and reporting weaknesses:

Namely, the manuscript does not provide sufficient detail on the search strategy. Merely listing the databases is insufficient; the exact keywords, Boolean operators, and search strings for each database must be explicitly reported. Therefore, I strongly recommend adding the full search strategy, either in the main text or in a supplementary table.

 We thank the reviewer for this valuable comment. We have revised the Methods section to provide a clearer and more detailed description of the search strategy as follow:

Search strategies were developed in collaboration with a university research librarian and included both controlled vocabulary (e.g., MeSH terms) and free-text keywords, adapted to the syntax and indexing of each database. The main search concepts combined terms related to social connectedness (“social isolation,” “loneliness,” “social participation,” “social interaction”) and stoma- or incontinence-related conditions (“ostomy,” “surgical stoma,” “urinary incontinence,” “fecal incontinence”). Boolean operators (AND/OR) were used to combine these concepts appropriately, ensuring sensitivity and specificity across databases (Table S1).

The full search strategy is reported in Supplementary file 1.

 Additionally, there is no mention of a pre-registered protocol (e.g., OSF). While not mandatory for scoping reviews, protocol registration is increasingly recognised as best practice. If the protocol was not registered, this should be clearly acknowledged as a limitation.

We appreciate this important observation. The protocol for this scoping review was not pre-registered. Accordingly, we have explicitly acknowledged this in the revised manuscript as a limitation as follows:

Finally, the review protocol was not pre-registered on a public repository. While registration is not mandatory for scoping reviews, it is increasingly recognized as good practice to enhance methodological transparency and reproducibility.

Furthermore, the authors state that they followed the PRISMA-ScR guidelines, but several checklist elements are missing or underdeveloped. For example, the inclusion and exclusion criteria are not fully detailed, and the reasons for exclusion at each stage of the screening process are not provided. These details should be systematically reported to align with PRISMA-ScR standards.

We thank the reviewer for this important observation. we have expanded and clarified the inclusion and exclusion criteria in the Methods section together with the main reasons for exclusion (in the results section) as follows:

Eligible studies included primary research and review articles employing qualitative, quantitative, or mixed-methods designs that explored aspects of social isolation, loneliness, social participation, or social interaction in individuals living with an ostomy, urinary incontinence, or fecal incontinence. Studies were included if they involved adult participants (≥18 years), explicitly addressed social or psychosocial dimensions related to the target conditions, were published in English or Italian, and were peer-reviewed full-text articles.

Grey literature such as conference abstracts, dissertations, editorials, and commentaries was excluded due to the absence of peer review and the difficulty in assessing methodological quality. Studies focusing exclusively on clinical or surgical outcomes without reference to social or psychological aspects, as well as non-human studies, were also excluded.

 …..

The database searches yielded a total of 314 records. After removing 26 duplicates and 36 records excluded for other reasons (e.g., non-research material or unavailable language), 252 records remained for screening. During title and abstract screening, 149 records were excluded as not pertinent to the research question.

A total of 103 potentially eligible records were sought for full-text retrieval. Four of these could not be accessed despite assistance from the institutional library service and attempts to contact the corresponding authors. Consequently, 99 full-text articles were assessed for eligibility. Of these, 80 were excluded for not meeting the predefined inclusion criteria, specifically, 43 for not addressing the target concept, 9 for non-eligible population, and 28 for aims unrelated to the review question.

Finally, I strongly recommend including a comprehensive table summarising the characteristics of the included studies (author, year, country, design, sample size, instruments used, and key findings). This would greatly facilitate visibility and allow readers to navigate more easily, which is a common practice in scoping review manuscripts.

We thank the reviewer for this helpful suggestion. In response, we have added a comprehensive data extraction table summarizing the key characteristics of all included studies, including author, year, country, study design, sample size, instruments used, and main findings. This table has been included in the text.

Reviewer 2 Report

Comments and Suggestions for Authors

The authors present a scoping review addressing the highly relevant topic of social isolation among individuals with incontinence or an ostomy. The study is timely and addressed a critical need in chronic illness management. The review was conducted systematically, and the synthesis of predisposing and precipitating factors is a valuable contribution to the field. The manuscript has considerable merit but requires revisions to strengthen its theoretical foundation, methodological transparency, and practical implications before it can be considered for publication.

1.The Middle Range Theory of Social Isolation in Chronic Illness is central to this review but is currently only introduced in the Discussion section. To provide a stronger conceptual foundation and guide the analysis, this theory should be positioned as the cornerstone of the study from the introduction. The research questions should be explicitly derived from or linked to the constructs of this theory. I recommend integrating the theory into the introduction and using it to frame the entire review. Besides, research questions should be explicitly generated based on this theory.

2.The review was restricted to literatures in English and Italian, which warrants some justification.

3.While Figure 1 provides a helpful conceptual model, it would greatly enhance the manuscript's clarity and rigor to include a summary table of the included studies. This table may contain detailed key information, e.g., author(s), year, country, study design, sample size, population characteristics, and key findings.

4.The discussion on the implications of applying the Middle Range Theory to incontinence studies is currently underdeveloped (lines 442-452). This section should be substantially expanded. For example, there are a series of propositions, patterns and mediational mechanisms in the theory, which may be guides for future studies.

Author Response

The authors present a scoping review addressing the highly relevant topic of social isolation among individuals with incontinence or an ostomy. The study is timely and addressed a critical need in chronic illness management. The review was conducted systematically, and the synthesis of predisposing and precipitating factors is a valuable contribution to the field. The manuscript has considerable merit but requires revisions to strengthen its theoretical foundation, methodological transparency, and practical implications before it can be considered for publication.

Thank you.

1.The Middle Range Theory of Social Isolation in Chronic Illness is central to this review but is currently only introduced in the Discussion section. To provide a stronger conceptual foundation and guide the analysis, this theory should be positioned as the cornerstone of the study from the introduction. The research questions should be explicitly derived from or linked to the constructs of this theory. I recommend integrating the theory into the introduction and using it to frame the entire review. Besides, research questions should be explicitly generated based on this theory.

 Thank you. Accordingly, we have revised the Introduction to integrate this theoretical model as the guiding framework underpinning the study rationale and objectives as follows:

This review is guided by the Middle Range Theory of Social Isolation in Chronic Illness [18], which provides a framework for understanding how chronic conditions contribute to or are influenced by social disconnection. The model differentiates between predisposing factors, reflecting stable sociodemographic and psychosocial vulnerabilities, and precipitating factors, referring to illness-related events that may trigger or worsen isolation. The outcomes encompass psychological, social, and health-related dimensions, capturing the wide-ranging effects of isolation on individual well-being. Guided by this theoretical framework, the present review was designed to explore how these constructs have been investigated in the context of incontinence and ostomy.

2.The review was restricted to literatures in English and Italian, which warrants some justification.

We thank the reviewer for this observation. The language restriction to English and Italian was applied for one main reason. These are the languages fluently spoken by all members of the research team, which ensured accurate screening, data extraction, and interpretation of findings without translation bias. We have now clarified this rationale in the Limitations section of the revised manuscript as follows:

First, only studies published in English or Italian were included, which may have led to the exclusion of relevant evidence published in other languages. This restriction was applied because these are the languages spoken by all members of the research team, allowing accurate interpretation of the data without the risk of translation errors.

3.While Figure 1 provides a helpful conceptual model, it would greatly enhance the manuscript's clarity and rigor to include a summary table of the included studies. This table may contain detailed key information, e.g., author(s), year, country, study design, sample size, population characteristics, and key findings.

We thank the reviewer for this valuable suggestion. In response, we have created a comprehensive summary table in the text including the main characteristics of all studies included in the review, such as author(s), year, country, study design, sample size, population characteristics, and key findings.

4.The discussion on the implications of applying the Middle Range Theory to incontinence studies is currently underdeveloped (lines 442-452). This section should be substantially expanded. For example, there are a series of propositions, patterns and mediational mechanisms in the theory, which may be guides for future studies.

We sincerely thank the reviewer for this insightful comment. We expanded discussion under the “Implications for future research” following the reviewer’s suggestion as follows:

From a research perspective, this review supports the use of the Middle Range Theory of Social Isolation in Chronic Illness as a conceptual framework to interpret the experiences of individuals with incontinence. This theory can guide future hypothesis-driven investigations by clarifying the mechanisms through which predisposing and precipitating factors lead to isolation and by identifying mediating processes such as self-concept erosion and loss of perceived control.

Reviewer 3 Report

Comments and Suggestions for Authors

Manuscript is interesting and touches important area of psychological aspect of incontinent people.

Here are my suggestions concerning the article:

1/ incontinence patients are not homogenous group - it is known that OAB patients tend to avoid social life more than SUI patients - can you provide informations about social isolation in different groups?

2/ as all groups of incontinence patients of all ages were assessed together, it is hard to generalize and draw exact conclusions, especially when it concerns at least 20% of population - maybe it is worth at least divide them into ostomy group or urine vs fecal incontinence group?

or to focus on one group of patients in the article?

3/ what was final number of patients included in the study?

Author Response

Manuscript is interesting and touches important area of psychological aspect of incontinent people.

Here are my suggestions concerning the article:

1/ incontinence patients are not homogenous group - it is known that OAB patients tend to avoid social life more than SUI patients - can you provide informations about social isolation in different groups? 2/ as all groups of incontinence patients of all ages were assessed together, it is hard to generalize and draw exact conclusions, especially when it concerns at least 20% of population - maybe it is worth at least divide them into ostomy group or urine vs fecal incontinence group? or to focus on one group of patients in the article?

We thank the reviewer for these insightful comments. We addressed reviewer questions 1 and 2 together. We fully agree that incontinence is a heterogeneous condition and that psychosocial experiences may differ across subtypes and conditions. However, the available evidence did not allow for subgroup comparisons. None of the included studies differentiated between specific types of urinary incontinence (e.g., overactive bladder vs stress urinary incontinence), and only four studies specifically examined individuals with an ostomy in relation to social isolation, all using qualitative methods and reporting similar experiential patterns. To address this we have expanded the Limitations section as follows:

Third, the heterogeneity of the included populations represents a key limitation of this review. Most studies focused on urinary incontinence, while evidence concerning ostomy remains scarce and largely qualitative. Furthermore, none of the studies distinguished between urinary incontinence subtypes (e.g., overactive bladder vs stress urinary incontinence), preventing more detailed comparisons. This imbalance restricts generalizability and precludes analysis of condition-specific mechanisms. Future research should examine these subgroups separately to capture differential psychosocial pathways and tailor interventions accordingly. Finally, the review protocol was not pre-registered on a public repository. While registration is not mandatory for scoping reviews, it is increasingly recognized as good practice to enhance methodological transparency and reproducibility.

3/ what was final number of patients included in the study?

We thank the reviewer for this question. As this work is a scoping review, the included studies were highly heterogeneous in design, population, and methodology; therefore, it would not be methodologically appropriate to calculate or report a pooled number of participants. Instead, we provided the sample size ranges to illustrate the variability across study designs. Specifically, qualitative studies included between 11 and 33 participants, whereas quantitative studies enrolled from 303 to 16,369 participants.

Round 2

Reviewer 1 Report

Comments and Suggestions for Authors

Dear Authors,

Thank you for your careful and thoughtful revisions to the manuscript. I appreciate your efforts in addressing my comments, which have significantly improved the overall quality of your manuscript.

Following the suggested recommendations, you have successfully clarified the search strategy, pre-registered protocol (e.g., OSF), and the inclusion and exclusion criteria in the Methods section, which adhere to the PRISMA-ScR guidelines. Additionally, you include a comprehensive table summarising the characteristics of the included studies (author, year, country, design, sample size, instruments used, and key findings).

Therefore, I am pleased to recommend this revised version for publication in Nursing Reports.

Sincerely,

Reviewer

Author Response

Dear Authors,

Thank you for your careful and thoughtful revisions to the manuscript. I appreciate your efforts in addressing my comments, which have significantly improved the overall quality of your manuscript.

Following the suggested recommendations, you have successfully clarified the search strategy, pre-registered protocol (e.g., OSF), and the inclusion and exclusion criteria in the Methods section, which adhere to the PRISMA-ScR guidelines. Additionally, you include a comprehensive table summarising the characteristics of the included studies (author, year, country, design, sample size, instruments used, and key findings).

Therefore, I am pleased to recommend this revised version for publication in Nursing Reports.

Sincerely,

Reviewer

Thank you. We truly appreciate your valuable comments and suggestions

Reviewer 2 Report

Comments and Suggestions for Authors

The authors have revised the manuscript as suggested. I have no further comments.

Author Response

The authors have revised the manuscript as suggested. I have no further comments.

Thank you. We truly appreciate your valuable comments and suggestions

  Reviewer 3 Report

Comments and Suggestions for Authors

I think Table 2 was a good idea. Also, the limitations part with information about lack of differentiation in the studies between OAB and SUI was important.

You mentioned in abstract about antecedents, but it is not clearly written in manuscript.

Also, important information is shown in Fisher study - that social isolation is concerning also parents or caretakers of incontinent patients, it is worth to highlight that in your work.

Author Response

I think Table 2 was a good idea. Also, the limitations part with information about lack of differentiation in the studies between OAB and SUI was important.

Thank you. We truly appreciate your valuable comments and suggestions

 You mentioned in abstract about antecedents, but it is not clearly written in manuscript.

Thank you for your valuable observation.
We have clarified and explained this aspect in the “Operationalization of the Concepts” paragraph al follow:

Within this framework, antecedents are distinguished as predisposing factors, representing stable sociodemographic and psychosocial characteristics that shape baseline vulnerability, and precipitating factors, denoting illness-related conditions or events that can initiate or intensify social isolation.

Also, important information is shown in Fisher study - that social isolation is concerning also parents or caretakers of incontinent patients, it is worth to highlight that in your work.

We have integrated Fisher’s finding, noting that social isolation may also affect parents and caregivers of incontinent patients. This aspect has been added in both the Results and Discussion sections as follow:

One study [46] also suggested that social isolation may extend to parents and caregivers, who can experience emotional strain and withdrawal related to the burden of care.

[…]

 Moreover, it is important to acknowledge that social isolation does not affect only individuals living with incontinence or those with an ostomy, but also extends to their close relatives and caregivers. Fischer’s study [46]suggests that social isolation may al-so affect parents and caregivers of incontinent patients, who can experience emotional strain and social withdrawal related to the burden of care. This finding points to the relational nature of isolation and highlights the need for further research on caregivers’ experiences.